# *Dioscorea nipponica* Makino Rhizome Extract and Its Active Compound Dioscin Protect against Neuroinflammation and Scopolamine-Induced Memory Deficits

**DOI:** 10.3390/ijms23179923

**Published:** 2022-09-01

**Authors:** Shofiul Azam, Yon-Suk Kim, Md. Jakaria, Ye-Ji Yu, Jae-Yong Ahn, In-Su Kim, Dong-Kug Choi

**Affiliations:** 1BK21 Program, Department of Applied Life Sciences, Graduate School, Konkuk University, Chungju 27478, Korea; 2BKplus GLOCAL Education Program of Nutraceuticals Development, Konkuk University, Chungju 27478, Korea; 3Melbourne Dementia Research Centre, The Florey Institute of Neuroscience and Mental Health, The University of Melbourne, Parkville, VIC 3052, Australia; 4Department of Biotechnology, College of Biomedical and Health Science, Research Institute of Inflammatory Disease (RID), Konkuk University, Chungju 27478, Korea

**Keywords:** *Dioscorea nipponica* Makino, dioscin, anti-inflammatory, LPS, neuroinflammation, cognitive deficits

## Abstract

Activation of microglial cells by intrinsic or extrinsic insult causes neuroinflammation, a common phenomenon in neurodegenerative diseases. Prevention of neuroinflammation may ameliorate many neurodegenerative disease progressions. *Dioscorea nipponica* Makino (DN) extract can alleviate muscular atrophy and inflammatory diseases; however, the efficacy and mechanism of action in microglial cells remain unknown. The current study investigates the possible anti-inflammatory effects and mechanisms of *Dioscorea nipponica* Makino ethanol extract and its steroidal saponin dioscin. Our in vitro study shows that *Dioscorea nipponica* rhizome ethanol extract (DNRE) and dioscin protect against lipopolysaccharide (LPS)-activated inflammatory responses in BV-2 microglial cells by inhibiting phosphorylation and the nuclear translocation of nuclear factor kappa-light-chain-enhancer of activated B cells (NF-κB), resulting in the downregulation of pro-inflammatory cytokines and enzymes. Consistent with our previous report of dioscin-mediated enhancement of neurotrophic factors in dopaminergic cells, here we found that dioscin upregulates brain-derived neurotrophic factor (BDNF) and cAMP-response element binding protein (CREB) phosphorylation (pCREB) in the cerebral cortex and hippocampus regions of the mouse brain. Scopolamine treatment increased pro-inflammatory enzyme levels and reduced the expression of BDNF and pCREB in the hippocampus and cortex regions, which led to impaired learning and referencing memory in mice. Pre-treatment of dioscin for 7 days substantially enhanced mice performances in maze studies, indicating amelioration in cognitive deficits. In conclusion, DNRE and its active compound dioscin protect against neurotoxicity most likely by suppressing NF-κB phosphorylation and upregulating neurotrophic factor BDNF.

## 1. Introduction

Neuroinflammation is a vital pathological feature of neurodegenerative disease progression, including Alzheimer’s disease [1,2]. Brain-residing macrophages, the microglia, can sense exogenous neurotoxins such as lipopolysaccharide (LPS)—the outer membrane polysaccharide component of Gram-negative bacteria). LPS binds to toll-like receptor 4 (TLR4) in the microglia, which stimulates an innate immune response [2,3,4]. Following TLR4 signalling activation, a downstream signalling cascade activates nuclear factor kappa-B (NF-κB), and this triggers the release of pro-inflammatory cytokines, including tumour necrosis factor-α (TNF-α), interleukin-1β (IL-1β), and interleukin-6 (IL-6) [5,6]. The inducible pro-inflammatory mediator isoforms of nitric oxide synthase (iNOS) and cyclooxygenase-2 (COX-2) are also upregulated in this process, and increase the production of nitric oxide (NO) and prostaglandin E2 (PGE2), which further exaggerate neuroinflammation [7].

The forebrain cholinergic neurons play an important role in controlling the central-nervous-system-regulated cognitive functions [8]. It has been seen that cholinergic decline impacts cognitive functions and disrupts anti-inflammatory pathways. Cholinergic signalling has been seen to regulate peripheral cytokine productions via anti-inflammatory mechanisms [9]. Scopolamine is an anticholinergic drug that impairs cognitive functions and learning memory, which resembles molecular and behavioural features of Alzheimer’s disease by impairing cholinergic neurotransmission in the hippocampus and prefrontal cortex [10,11]. In addition, these neuropsychiatric changes also involve the upregulation of pro-inflammatory cytokines and inflammasome components in different brain regions [12].

*Dioscorea nipponica* (DN), a species of the Dioscorea genus belonging to the family Dioscoreaceae, is an important herb in Chinese traditional medicine [13]. It has commonly been prescribed to treat respiratory illnesses such as dry cough and bronchitis, rheumatism and analgesia, stimulate blood circulation, and enhance digestion and diuresis [13,14]. Major compounds found in *Dioscorea nipponica* are steroidal saponins, such as dioscin [13], and stilbene derivatives, such as diosniponol C and diosniposide A [15]. Recent findings have shown that bioactive compounds of *Dioscorea nipponica*, and steroidal saponin, attenuate cardiovascular diseases, improve immune functions, and prevent cancer progression [13]. *Dioscorea nipponica* has also been shown to promote muscle differentiation, restore muscle atrophy, and recover from injury by inhibiting the NF-κB pathway [16]. The total saponin fraction and phenolic compounds from *Dioscorea nipponica* extract have been shown as potential modulators of the MAPK signalling pathway in an inflammatory disease model [15,17]. However, the biological impact of *Dioscorea nipponica* in neuroinflammation progression or prevention is largely unknown. Therefore, this study investigates the possible anti-inflammatory effect of *Dioscorea nipponica* rhizome ethanol extract (DNRE) on microglial cell activation and pro-inflammatory cytokine production.

## 2. Results

### 2.1. DNRE Protects Microglial Cells from LPS-Induced Stress

To test the hypothesis that DNRE treatment could ameliorate inflammatory cell death and reduce NO release from microglia, we tested different doses of DNRE (10–400 µg/mL). The treatment with DNRE was shown to have a dose-dependent effect in reducing NO release, but high doses of DNRE (200–400 µg/mL) showed toxicity (Figure 1A). Thus, considering the non-toxic dose and IC_50_ of MTT assay (152.2997 µg/mL), the better effect was attributed to the dose of 50–100 µg/mL (Figure 1A,B). Therefore, we chose 10–100 µg/mL for the rest of the study. This effect might be related to the antioxidant effect of DNRE (0.01–1.0 mg/mL), which was supported by the DPPH scavenging assay and LPS-induced ROS release by DCFDA assay (Figure 1C,D).

### 2.2. DNRE Pre-Treatment Suppresses Inflammatory Mediators following LPS Toxication

In response to the microglial activation, pro-inflammatory mediator iNOS and COX-2 levels also increase via a mechanism involving increased NO and PGE2 level [7]. Several studies reported that systemic injection of LPS triggers neuroinflammation, leading to neurodegeneration and cognitive dysfunction [18,19,20]. As we have seen, DNRE treatment significantly reduced NO release and scavenged DPPH radicals; therefore, we investigated DNRE’s impact on post-LPS-treated inflammatory biomarkers. Pre-treatment of DNRE showed a dose-dependent effect in reducing inflammatory enzymes COX-2 and iNOS mRNA and protein levels (Figure 2A,B). DNRE pre-treatment was shown to reduce, by at least twofold, iNOS mRNA or protein level (Figure 2A,B), signifying DNRE-treated decrease in NO release. This result was further justified by the reduction in mRNA level of IL-1β and IL-6 at least twofold in LPS-induced BV-2 microglial cells (Figure 2C). DNRE also significantly downregulated LPS-induced increase inflammatory cytokine TNF-α production in BV-2 microglial cells (Figure 2C).

### 2.3. DNRE Inhibits Nuclear Translocation of NF-κB

LPS-induced microglial activation hyperactivates the MAPK pathway and exaggerates inflammation by promoting transcription factors such as NF-κB [4]. DNRE treatment dose-dependently decreased phosphorylation of MAPK signalling by reducing p-ERK, p-JNK, and p-p38 (Appendix A) in LPS-treated BV-2 cells.

Phosphorylation of the p65 subunit of NF-κB translocates into the nucleus to translate inflammatory cytokines such as IL-1β, and causes inflammation. As we found that DNRE reduced phosphorylation of several biomarkers of MAPK signalling, we investigated whether DNRE treatment inhibited nuclear translocation of transcription factor NF-κB. LPS induction activates NF-κB by phosphorylating IκB at the cytosol, promoting the p65 subunit of NF-κB translocation into the nucleus [4]. DNRE treatment (100 µg/mL) was shown to inhibit IκB phosphorylation (Figure 3A) and reduce p65 permeation into the nucleus (Figure 3B). To confirm this finding, we used fluorescence microscopy to visualise DNRE impact on phosphorylated p65 subunit, and found a similar decrease to that observed with the WB probe (Appendix A).

### 2.4. Anti-Neuroinflammatory Activities of Dioscin

Multiple studies have indicated that DNRE consists of steroidal saponin derivatives; dioscin is the most abundant among them [21]. In addition, our recent study found that dioscin ameliorates MPP^+^-induced neurotoxicity via autophagy [22]. We observed that dioscin substantially downregulates inflammatory enzymes iNOS and COX-2 in LPS-activated BV-2 cells. Thus, we used HPLC fingerprinting to identify and quantify dioscin in DNRE (Figure 4A). Comparing the retention times with standard dioscin (Figure 4A top), we confirmed the identity of dioscin in DNRE (Figure 4A bottom). Subsequently, we calculated the concentrations of dioscin using peak area, and found that 7.07 ± 2.95 µg of dioscin is present per milligram of our ethanolic crude extract of DN. Interesting unknown peaks were observed between retention time 5–10 min, before dioscin peaks, in the DNRE chromatogram. We speculate that these could be other saponin constituents of DNRE; further analysis should be performed to identify these constituents.

Since DNRE-mediated neuroprotection partly relies on its potential antioxidant effects, we tested whether dioscin possesses an antioxidant effect. In the ABTS analysis, dioscin showed more efficiency in free radicals scavenging than DNRE (Figure 4B). Previously, we showed that dioscin is a potential neuroprotector in an MPP^+^-induced Parkinson’s disease model [22], where dioscin showed a substantial reduction of apoptosis and upregulated neurotrophic factors such as BDNF and pCREB. Additionally, dioscin ameliorated inflammatory enzyme upregulation in LPS-stimulated in vitro model. Thus, we tested the speculation that dioscin could ameliorate phosphorylation and nuclear translocation of NF-κB p65 subunits (Figure 4C). We used two doses of dioscin and found that both doses are highly capable of downregulating p65′s redistribution. Since NF-κB p65 phosphorylation and nuclear localisation could transcribe multiple inflammatory cytokines, one of the therapeutic strategies is to prevent p65 nuclear translocation [23]. Our data suggest that the dioscin of DNRE could be an inhibitor of p65 phosphorylation and translocation, which could prevent neuroinflammation.

### 2.5. Dioscin Ameliorates Scopolamine-Induced Learning Deficits

Based on our previous finding of dioscin-induced improvement on neurotrophic factors in neuronal cells [22], we tested dioscin’s impact on the scopolamine-induced amnesic mice model. Scopolamine is an anticholinergic drug that antagonises cholinergic receptors and impairs hippocampal and prefrontal cortex transmission that deficits learning acquisition [10]. Intraperitoneal injection of scopolamine also stimulates systemic and consequent neuroinflammation in C57BL/6 mice [12]. Dioscin (60 mg/kg) treatment for 7 days ameliorated scopolamine-induced deficits in spatial learning memory in mice (Figure 5A–C).

### 2.6. Dioscin Ameliorates Scopolamine-Induced Neurotrophic Factor Deficits

Scopolamine treatment significantly upregulated pro-inflammatory enzyme iNOS and COX-2 expression (Figure 6A) in the cerebral cortex and hippocampus, which was significantly reduced by dioscin treatment. In addition, dioscin treatment also significantly improved neurotrophic factors BDNF and pCREB expressions (Figure 6B). These results indicate that dioscin ameliorates pro-inflammatory enzymes and neurotrophic factors induced by scopolamine, and improves spatial learning memory. Our findings also suggest that the dioscin might have a better effect on the cerebral cortex region than on the hippocampus; however, further investigation needs to be conducted to understand region-specific effects.

## 3. Discussion

Microglia are tissues residing macrophages that maintain the neuronal microenvironment and respond to endogenous and/or exogenous injury. Unlike peripheral macrophages, microglia possess phenotype-based polarisation features, which allow them to function as innate immune system modulators [24]. LPS can stimulate the microglial M1 phenotype and induce an inflammatory response via increasing cytokine release. This study hypothesised that DNRE could ameliorate LPS-induced microglial shifting and downregulate inflammatory response. DNRE has substantially decreased NO release and phosphorylation of different downstream effectors of MAPK signalling in LPS-activated BV-2 microglia.

LPS activates microglia and releases pro-inflammatory enzymes and cytokines, such as iNOS, IL-1β, IL-6, and TNF-α [25]. These are the key mediators of acute and chronic inflammation, leading to neurodegenerative diseases. In this study, LPS treatment substantially increased NO release and, subsequently, mRNA level of iNOS, IL-1β, IL-6, and TNF-α, and protein level of iNOS in BV-2 microglial cells. This indicates that DNRE pre-treatment might have reversed the M1 polarisation and reduced pro-inflammatory cytokines mRNA and protein expression. COX-2 is another pro-inflammatory enzyme synthesising pro-inflammatory prostaglandin E2 (PGE2) when exposed to LPS [26]. Consistently, LPS treatment significantly increased both COX-2 mRNA and protein levels that were alleviated by DNRE pre-treatment. Thus, we assume that DNRE pre-treatment could inactivate the NF-κB phosphorylation and MAPK signal transduction pathways (Appendix A) responsible for pro-inflammatory cytokine production [17,27,28].

Several previous studies have published that DNRE contains bioactive compounds, including phenols and saponins [15,29], and these compounds might have been responsible for anti-inflammatory activity in this study. Among the several compounds, dioscin has been found to act as a primary bioactive compound in DNRE [29], which has shown anti-inflammatory activity in several independent studies [30,31]. In this study, HPLC fingerprinting assay found substantial levels of dioscin in DNRE. Dioscin also showed significant inhibition of NF-κB (p65) nuclear translocation, suggesting that DNRE-mediated inhibition of neuroinflammation could be associated with the presence of dioscin.

Our previous study [22] showed that dioscin is a potential bioactive compound that enhances neurotrophic factors in neuronal cells. In corroboration to prior findings, we found that dioscin treatment improves spatial and reference memory in scopolamine-treated animal models. MWM is not a maze in the usual sense–a labyrinth; instead, it uses hippocampal synaptic plasticity and NMDA receptor function [32]. We found that dioscin treatment improved mice performance in the search-to-platform area and avoided latency, which indicated improvement in synaptic plasticity. BDNF and other neurotrophic factors regulate the neural plasticity network [33]; dioscin treatment significantly upregulated neurotrophic factors in the hippocampus and cortex region. Our data validate the mice performances in MWM and Y-maze analyses.

Scopolamine inhibits cholinergic transmission in the hippocampus and prefrontal cortex region, affecting spatial and related learning memory. In this process of neurocognitive impairments, scopolamine increases inflammatory cytokines TNF-α, IL-6 and IL-1β [12], and enzymes iNOS and COX-2 [34]. In the animal model of scopolamine, dioscin treatment for 7 days reduced pro-inflammatory mediators, which follows the inhibition of NF-κB phosphorylation. In agreement with our previous reports, this study found that dioscin provides neuroprotection possibly via the upregulation of neurotrophic factors in different brain regions. This study provides a scientific basis for ethnobotanical use of DNRE and suggests further investigation needs to be conducted with its potential active compound, dioscin, emphasising the regulation of neurotrophic factors.

## 4. Materials and Methods

### 4.1. Chemicals

DNRE was purchased from Korea Plant Extract Bank (ID: PBC-464AS), lipopolysaccharide (LPS; *Escherichia coli*; 055:B5), dimethyl sulfoxide (DMSO), 3-(3,4-dimehylthiazol-2-yl)-2,5-diphenyltetrazolium bromide (MTT), and scopolamine hydrochloride were obtained from Sigma-Aldrich (St. Louis, MO, USA). Foetal bovine serum (FBS) (#16000-442; Gibco, NY14072, USA), phosphate-buffered saline (PBS), and Dulbecco’s modified Eagle’s medium (DMEM) were purchased from Gibco-BRL Technologies (Gaithersburg, MD, USA). Trizol was purchased from Invitrogen Life Technologies (Carlsbad, CA, USA). RIPA buffer (10x) was purchased from Millipore (Milford, MA, USA), and protease and phosphatase inhibitors were obtained from Roche (Indianapolis, IN, USA).

### 4.2. Cell Culture and Treatment

The BV-2 microglial cells were a generous gift from Dr K. Suk (Kyung-Pook National University, South Korea) [35]. Cells were grown in DMEM supplemented with 5% (*v*/*v*) FBS and 1% (*v*/*v*) of penicillin/streptomycin in a maintained incubator (37 °C and 5% CO_2_). Upon reaching confluency (80–90%), cells were trypsinised (0.05% trypsin-EDTA) for sub-culture and/or for seeding before respective treatment. For each experiment, cells from at least three consecutive passages were used. The BV-2 cells were exposed to different doses of DNRE (10–400 µg/mL) and dioscin (200 and 400 ng/mL) for 2 h before being exposed to LPS (200 ng/mL) for 24 h.

### 4.3. Cell Viability and Nitric Oxide Assay

For assessing cell viability and nitric oxide (NO) release, the BV-2 cells were seeded at 5 × 10^3^ cells/well in a 96-well plate. Upon reaching approx. 80% confluency, cells were treated with various doses of DNRE (10–400 µg/mL) with or without LPS (200 ng/mL), and incubated for 24 h. Subsequently, the culture medium was collected for colourimetric NO release assay using Griess reagent; measurements were detected at 540 nm on the UV spectrum using a microplate reader (SunriseTM, Tecan Trading AG, Switzerland).

For cell viability, 20 µL of MTT (5 mg/mL) was added and incubated for 45 min. Following this, culture media was removed, and 200 µL of DMSO was added to dissolve the formazan crystals. Absorption was measured at 552 nm using a microplate reader.

### 4.4. DPPH Free Radical Scavenging Assay

2,2-Diphenyl-1-picrylhydrazyl (DPPH; 95%) was purchased from Thermo Fisher Scientific (Ward Hill, MA 01835, USA). A working stock of DPPH (200 µM/L) was prepared in ethanol (99%), and was added to different concentrations of DNRE to make up a final volume of 300 µL/well in a 96-well plate. The first lane contained DPPH + ethanol as the control, and an additional lane was filled with ethanol alone as a blank (negative control). Absorbance was measured in a microplate reader at 520 nm. The calculation for DPPH radical inhibition was as follows:(1)$$\%\;of\;DPPH\;radical\;inhibition=\left[\frac{{Abs}_{control}-({Abs}_{sample}-{Abs}_{blank})}{{Abs}_{control}}\right]\times 100$$

### 4.5. DCFDA Assay

To detect the intracellular ROS release after LPS induction, we used ROS-sensitive fluorescent dye, 2′,7′-dichlorofluorescein diacetate (DCFDA; Sigma-Aldrich). BV-2 cells (2.5 × 10^5^ cells/mL) were cultured in 6-well plates in regular media. Upon reaching the confluency (60–70%), LPS (200 ng/mL) followed by DNRE (10–100 µg/mL) was treated for 2 h in a serum-free media. Later, treated or non-treated cells were incubated for an additional 30 min at 37 °C with DCFDA (10 μM final concentration) and then washed twice with ice-cold PBS. Images were immediately captured using a fluorescence microscopy system (Nikon Eclipse Ts2R). The fluorescence intensity was calculated using ImageJ (Java 1.8.0_172, NIH).

### 4.6. ABTS Radical Scavenging Activity

The total antioxidant activity of the DNRE and Dio was measured using the ABTS + radical cation decolourisation assay [36]. A 7.4 mM of ABTS was mixed in equal volume with 2.6 mM potassium persulfate. The mixture was then stored in the dark at room temperature (RT) for 12–14 h before use. After radical generation, the ABTS radical solution was diluted with deionised water until its absorbance was 0.70 ± 0.02 at 734 nm. Then 0.9 mL of ABTS radical solution was mixed with 0.1 mL of samples, and the absorbance was measured at 734 nm. The antioxidant activity of the DNRE and Dio were expressed as Trolox equivalents antioxidant capacity (TEAC) or mM of Trolox equivalent per mg of extract (mM Trolox eq./mg extract).

### 4.7. Nucleic–Cytosolic Fraction Preparation

BV-2 cells were seeded into a 60 mm cell culture Petri dish at 2.5 × 10^5^ cells/mL and ~80% confluency; cells were treated with LPS or LPS + DNRE, or were not treated, for 24 h. Washed twice with ice-cold PBS, nucleic acid lysis buffer and 10% NP-40 (detergent) were used to collect cytosolic fractions at 4000 rpm for 10 min. Pellets were dissolved in extraction buffer (containing 20 mM HEPES, 20% glycerol and 0.2 mM EDTA) and incubated on ice for 30 min with occasional tapping to liberate the mixture to collect nucleic acid fractions. The mixture was centrifuged at 13,000 rpm for 15 min to collect nucleic acid fraction as supernatant, and both fractions were quantified and prepared for Western blot analysis.

### 4.8. Immunofluorescence

As described by Runwal et al. (2019) [37], treated and non-treated cells were washed with cold PBS once, followed by fixation with 4% cold PFA. Next, cells were permeabilised with 0.1% (*v*/*v*) Triton X-100 for 10 min at room temperature and washed (2x). Subsequently, primary antibody (anti-pNF-κB) at 2 µg/mL was added and incubated overnight at 4 °C. The next day, cells were counterstained with chicken anti-rabbit secondary antibodies (CAR-594; A21201 Invitrogen) at room temperature for 1 h. Finally, cells were stained with DAPI (2 µg/mL), and images were captured using a wide-angle fluorescence microscope, and were processed by NIS-Elements software (BR-2.01.00, NY 11747-3064, New York, USA).

### 4.9. Determination of a Possible Active Compound in the Extract

We performed high-performance liquid chromatography (HPLC) to determine and quantify a marker compound in DNRE [21,38,39]. A Thermo Scientific (Ultimate 3000) HPLC system (Thermo Scientific, Korea) equipped with a UV detector and 20 µL injection loop was used. We used a gradient system consisting of (A) 0.1% acetic acid in distilled water, and (B) 100% acetonitrile, and separated using Diamonsil (C18; 250 × 4.6 mm; 5 µm) column. The gradient ratio was set to 45:55 (A:B) for 30 min at the flow rate of 1.0 mL/min, and the UV detection was set to 210 nm. Using these conditions, we detected and quantified dioscin in the extract.

### 4.10. Animals Handling and Treatment

Male 8-week-old C57BL/6 (20–25 g) mice were purchased from Daehan Bio-Link, South Korea. All animals were housed in a controlled environment, as described by Jo and colleagues [40]. Five animals per group (control, scopolamine, and dioscin + scopolamine) were acclimatised for 1 week in different cages before experiments (Figure 7). Scopolamine (2 mg/kg) was administered intraperitoneally, and dioscin (60 mg/kg) and saline (control) were administered orally, for 7 consecutive days. All experiments were approved by the Institutional Animal Care and Use Committee (IACUC), Konkuk University (IACUC no- KUB201101).

### 4.11. Y-Maze Test

We used a Y-maze assay for spatial and reference memory assessment [41]. A ‘Y’ shaped maze with three identical arms, each arm separated at a 120° angle, was used for the test. Mice were placed at the centre of the maze, and latency to move and altered entry into different arms were noted for 5 min. Mice with intact memory should show less interest to re-visit recently entered arm, while those with impaired memory have high interest to re-visit.

### 4.12. Morris Water Maze Test

A Morris water maze (MWM) [32] evaluates the spatial memory of rodents, which relies on rodents’ navigation skills to find the submerged escape platform. This test used a circular pool (122 cm) with a hidden platform. The aim of the animal was to find a short and direct path to the hidden platform from the start position. We used two back-to-back training sessions with 2 h intervals for 2 days before treatment. Post-treatment, for the respective treatment plan, the spatial acquisition was measured and data analysed using SMART 3.0 software (Ver. 3.0, Harvard Apparatus, Holliston, MA 01746, USA).

### 4.13. Reverse Transcription–Polymerase Chain Reaction (RT–PCR)

BV-2 cells were washed twice with ice-cold PBS, and 1 mL of trizol reagent was added to each well and incubated at room temperature for 30 s. Cells were collected carefully to an Eppendorf tube followed by centrifugation. We used the ReverTra Ace-α kit (Toyobo, Osaka, Japan) to isolate first-strand cDNA, according to the manufacturer’s instructions. In addition, 1 µL of RT-mixture templet was used for further amplification in the presence of specific primers (Table 1). PCR products were electrophoresed in 1.5% agarose gel containing GelRed^®^ Nucleic Acid Gel Stain (Biotium Inc., Fremont, CA 94538, USA).

### 4.14. Western Blot Analysis

The cells were washed twice with ice-cold PBS and lysed using lysis buffer (1x RIPA lysis buffer containing a protease and phosphatase inhibitor (1:1) cocktail). Equal protein (20 µg/10 µL) were loaded to each lane for electrophoresis in ~8–15% sodium dodecyl sulphate-polyacrylamide electrophoresis (SDS-PAGE) gel. Electrophoresed proteins were then transferred onto polyvinylidene-difluoride (PVDF) membranes (Millipore, Bedford, MA, USA). The membranes were incubated at room temperature with 5% skim milk to prevent nonspecific binding. Then the blots were incubated overnight at 4 °C on a rocking platform with specific primary antibodies, including anti-iNOS (#A0312, ABclonal), anti-COX-2 (M-19 #sc1747, Santa Cruz biotechnology), anti-p38 (#9212S, cell signaling), anti-pp38 (#9201S, cell signaling), anti-JNK (#9253S, cell signaling), anti-pJNK (#9251S, cell signaling), anti-ERK (#91012S cell signaling), anti-pERK (#9101S, cell signaling), anti-NF-κB/p65 (F-6 #sc-8008, Santa Cruz biotechnology), anti-pNF-κB/pp65 (#3033S, cell signaling), anti-IκB (#4812S, cell signaling), anti-pIκB (#2859S, cell signaling), anti-nucleolin (#14574; cell signaling) at concentration of 1:1000, and anti-β-actin (#C4; Santa Cruz biotechnology) at concentration of 1:5000. The next day, each blot was incubated at room temperature with either an anti-mouse or anti-rabbit (1:10,000) secondary antibodies. The blots were visualised with an enhanced chemiluminescence detection system (LAS 500; GE Healthcare Bio-Sciences AB, 751 25, Uppsala, Sweden) per the recommended protocol.

### 4.15. Statistical Analysis

All statistical analyses were performed using GraphPad Prism (by Dotmatics, version-8.0.1; La Jolla, CA, USA) software. Data represent the mean ± SEM (standard error mean) of three independent experiments, at least. We used one-way ANOVAs followed by Sidak’s multiple comparisons to determine the statistical significance. The *p*-values were considered significant at <0.05.

## 5. Conclusions

DNRE pre-treatment significantly reversed LPS-mediated microglial activation, resulting in the downregulation of NF-κB (p65) phosphorylation and subsequent neuroinflammation. We found that dioscin, an active compound of DNRE, reduces p65 phosphorylation in vitro and upregulates neurotrophic factors in the hippocampus and cortex region of scopolamine-induced mice. Our data indicate that the DNRE-mediated amelioration of neuroinflammatory responses and memory deficits is partly attributed to the dioscin effect.

## Figures and Tables

**Figure 1 ijms-23-09923-f001:**
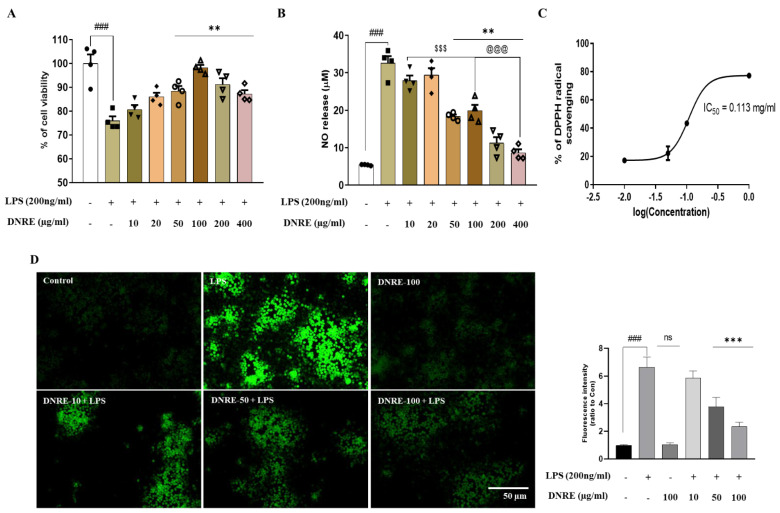
DNRE treatment protects microglial cells. (**A**) MTT assay (IC_50_ = 152.2997 µg/mL) and (**B**) nitric oxide release assay for LPS-treated BV-2 cells with or without different doses of DNRE. (**C**) Sigmoid plot for determination of IC_50_ value of DNRE of DPPH free radical scavenging. The half maximal inhibitory concentration (IC_50_) was calculated from the equation (y = mx-c) obtained from the sigmoid curve. The value of ‘y’ was changed to 50, and we were subjected to determine value of ‘x’, which is the IC_50_ value. (**D**) Fluorescence microscopy of DCFDA assay for LPS-treated BV-2 cells with or without DNRE, and measurement of fluorescence intensity using ImageJ software. ns—not significant compared with non-treated; each shape (circles, triangles, squares, etc.) is representative of the number of repetitions of a particular group; ### *p* < 0.001 compared with non-treated; ** *p* < 0.01, *** *p* < 0.001 comparing LPS vs. LPS + DNRE; ^$$$^
*p*< 0.001 comparing low-dose DNRE (10–20 µg/mL) vs. mid-to-high-dose DNRE (50–400 µg/mL), ^@@@^
*p* < 0.001 comparing mid-dose DNRE (50–100 µg/mL) vs. high-dose DNRE (200–400 µg/mL).

**Figure 2 ijms-23-09923-f002:**
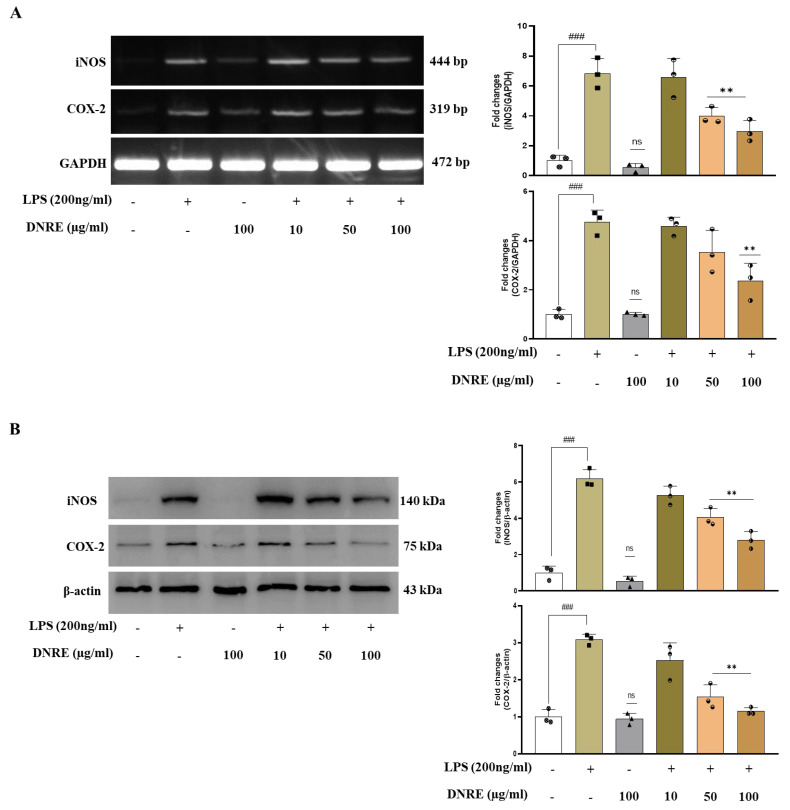
DNRE activities on inflammatory mediators. (**A**) mRNA and (**B**) protein expression of inflammatory enzymes in LPS (200 ng/mL)-treated BV-2 cells with or without different doses of DNRE. (**C**) mRNA expression of pro-inflammatory cytokines in LPS-treated BV-2 cells with or without DNRE. ns—not significant compared with non-treated; each shape (circles, triangles, squares, etc.) is representative of the number of repetitions of a particular group; ### *p* < 0.001 compared with non-treated; ** *p* < 0.01, *** *p* < 0.001 comparing LPS vs. LPS + DNRE.

**Figure 3 ijms-23-09923-f003:**
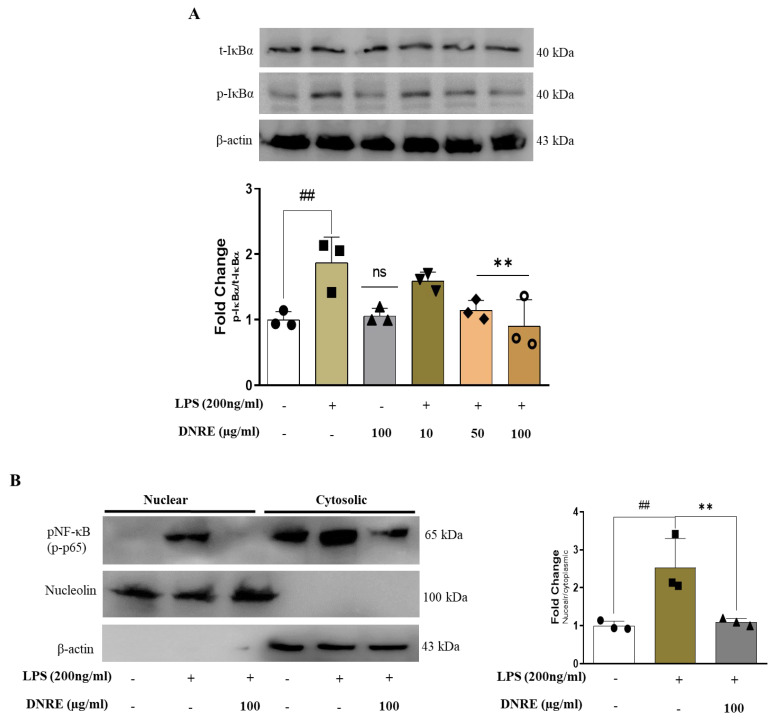
DNRE effects on nuclear factor. (**A**) Protein expression of IκB and densitometric calculation. (**B**) Protein expression of phosphorylated NF-κB (p-p65) of nuclear and cytoplasmic fractions of BV-2 cells treated with or without LPS (200 ng/mL). ns—not significant compared with non-treated; each shape (circles, triangles, squares, etc.) is representative of the number of repetitions of a particular group; ## *p* < 0.01 compared with non-treated; ** *p* < 0.01 comparing LPS vs. LPS + DNRE.

**Figure 4 ijms-23-09923-f004:**
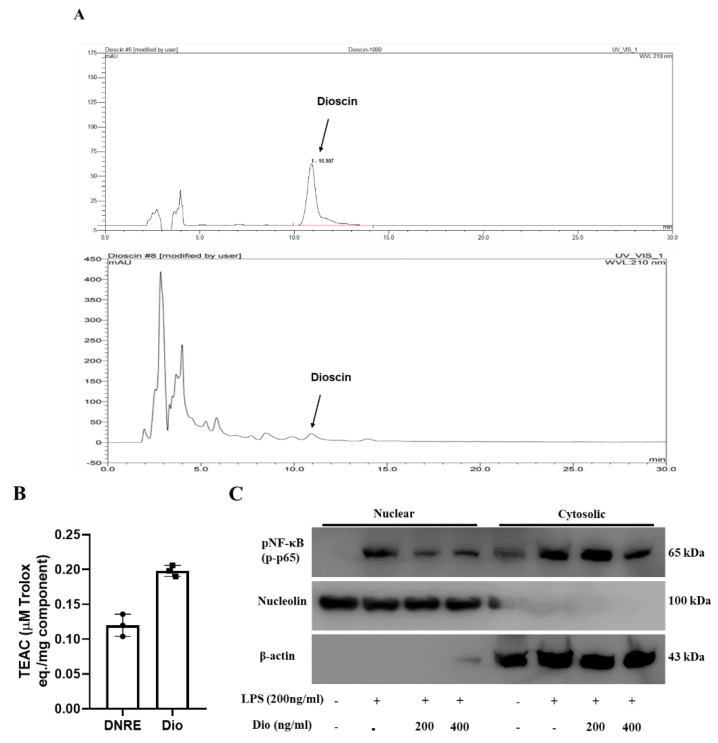
HPLC screening was conducted to identify the presence of and quantify dioscin in DNRE. (**A**) Representative chromatogram showing retention time and area of a standard steroidal saponin dioscin (**top**) and dioscin in DNRE (**bottom**). (**B**) The antioxidative capacity of dioscin (Dio) was measured using ABTS assay; an equimolar amount of DNRE and Dio was used, and the spectrophotometric measurements are expressed as µM of Trolox per mg of DNRE or Dio. (**C**) Nuclear and cytosolic fractions were isolated from LPS-activated BV-2 microglial cells pre-treated with or without different doses of dioscin and probed with anti-phosphorylated NF-κB (p-p65).

**Figure 5 ijms-23-09923-f005:**
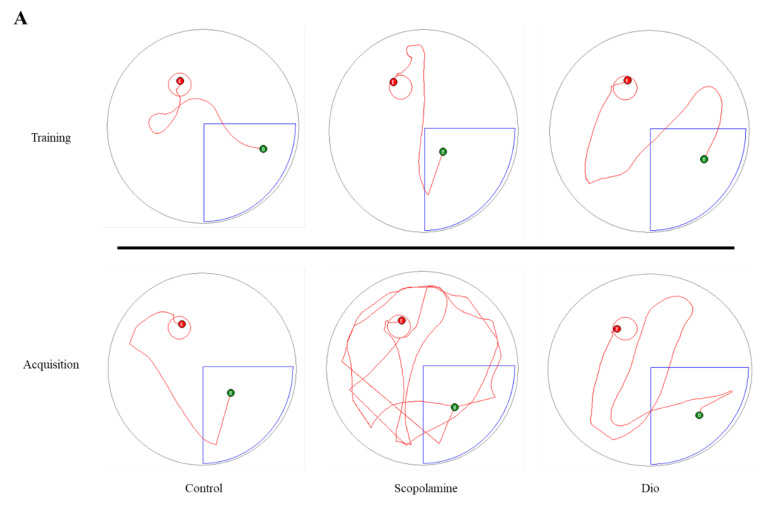
Dioscin effect on scopolamine-induced amnesic mice model. (**A**) Trajectory map used by animals during training and data acquisition in MWM pool; B = starting point and E = hidden platform. (**B**) Distance travelled by and latency to find a hidden platform in MWM; ns—not significant compared scopolamine vs. dioscin + scopolamine, ** *p* < 0.01 comparing scopolamine vs. dioscin + scopolamine. (**C**) Mice intensity of arms entry and alterations in Y-maze assay followed by treatment vs. non-treated.

**Figure 6 ijms-23-09923-f006:**
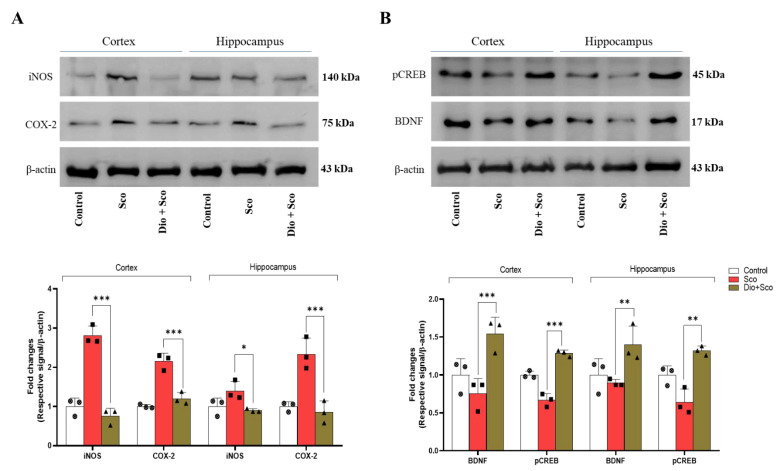
Dioscin effect on scopolamine-stimulated inflammatory response. (**A**) Dioscin treatment and pro-inflammatory response in cerebral cortex and hippocampus, and (**B**) neurotrophic factors. Sco, Scopolamine; Dio, Dioscin; each shape (circles, triangles, squares, etc.) is representative of the number of repetitions of a particular group; * *p* < 0.05, ** *p* < 0.01, and *** *p* < 0.001 comparing scopolamine vs. dioscin + scopolamine.

**Figure 7 ijms-23-09923-f007:**
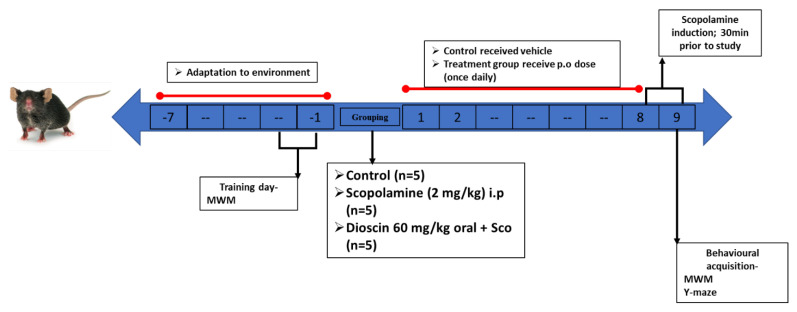
Schematic presentation of scheduling for animal grouping, treatment, behaviour, and sacrifice.

**Table 1 ijms-23-09923-t001:** Specific primer sequences used in this study.

Gene	Primer Sequence	Size (bp)
iNOS	F 5′-GAG GTA CTC AGC GTC CTC CA-3′ R 5′-AGG GAG GAA AGG GAG AGA GG-3′	444
COX-2	F 5′-TGA GTG GTA GCC AGC AAA GC-3′ R 5′-CTG CAG TCC AGG TTC AAT GG-3′	319
TNF-α	F 5′-TTC GAG TGA CAA GCC TGT AGC-3′ R 5′-AGA TTG ACC TCA GCG CTG AGT-3′	390
IL-1β	F 5′-CAA GGA GAA CCA AGC AAC GA-3′ R 5′-TTG GCC GAG GAC TAA GGA GT-3′	428
IL-6	F 5′-GGA GGC TTA ATT ACA CAT GTT-3′ R 5′-TGA TTT CAA GAT GAA TTG GAT-3′	435
GAPDH	F 5′- ACC ACA GTC CAT GCC ATC AC-3′ R 5′- CCA CCA CCC TGT TGC TGT AG-3′	472

## Data Availability

The datasets generated and/or analysed during the current study are publicly available upon acceptance of this manuscript.

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
