# Peer review of "Dioscorea nipponica Makino Rhizome Extract and Its Active Compound Dioscin Protect against Neuroinflammation and Scopolamine-Induced Memory Deficits"

_ijms, 2022, doi:10.3390/ijms23179923_

Round 1

Reviewer 1 Report

The study aimed to investigate the protective effects of Dioscorea nipponica rhizome ethanol extract (DNRE) and its major active compound dioscin protect against lipopolysaccharide (LPS)-activated inflammatory responses in neurons. In general, the authors have conducted thoroughly studies with in vitro and in vivo methods. The studies methods and the results have existed with completeness and readability. However, some of minor concerns were required to be examined for further improve the manuscript.

1.      Fig. 1B

Please make-up the statistical analysis data among the DNPE + LPS groups.

2.      In Fig. 5
The authors should present more data to indicate the quantitation data of dioscin in the DNRE. Moreover, the amount of dioscin seemed extremely low compared to the compounds shown before the retention time at 5 min. The questions are what are these compounds and how about their amounts. These data would affect the activity source of DNRE. And, the further discussion on the real active compounds of DNRE would be required.

Author Response

Title:               Dioscorea nipponica Makino rhizome extract and its active compound dioscin protect against neuroinflammation and scopolamine-induced memory deficits

Journal:          IJMS

Mans. ID        ijms-1823621

Editor

We thank you and reviewer for your and their important time and attention towards our submission. It was your valuable and insightful comments that led to possible improvements of our manuscript in the current version. All authors have considered comments raised by the reviewers and tried their best to address them to the point. We hope that our revised manuscript would meet your high standards. Below you could find authors point-by-point response to the reviewer’s comments. We have indicated all the changes in manuscript as “track change” for the understanding and quick finding.

Reviewer 1:

The study aimed to investigate the protective effects of Dioscorea nipponica rhizome ethanol extract (DNRE) and its major active compound dioscin protect against lipopolysaccharide (LPS)-activated inflammatory responses in neurons. In general, the authors have conducted thoroughly studies with in vitro and in vivo methods. The studies methods and the results have existed with completeness and readability. However, some of minor concerns were required to be examined for further improve the manuscript.

  1. 1B: Please make-up the statistical analysis data among the DNPE + LPS groups.

Response: Appreciate this concern, we have done statistical comparison in-between DNRE+LPS doses in Figure 2 B. We have described the statistical comparison in the figure legend as comparison between low vs mid vs high doses. In addition, we have calculated and added IC50 value of MTT assay in the figure legend section.

  1. In Fig. 5 The authors should present more data to indicate the quantitation data of dioscin in the DNRE. Moreover, the amount of dioscin seemed extremely low compared to the compounds shown before the retention time at 5 min. The questions are what are these compounds and how about their amounts. These data would affect the activity source of DNRE. And, the further discussion on the real active compounds of DNRE would be required.

Response: Appreciate this concern, Rhizome of D. nipponica has abundant steroidal saponins, and dioscin is one of the major compounds so far identified. We agree with reviewer and appreciate this critical insight, peaks appeared before 5 mins could be impurities or solvents, since we saw identical peaks in standard dioscin chromatogram too. However, concentrated peaks between 5 – 10 mins RT could be the other potential saponins of DNRE. In result section of the revised manuscript, we have addressed this concern properly.

We appreciate and understand that all these comments were raised to improve our manuscript's quality and readability. The authors have attempted their best to address all comments, and we welcome further enquiries by the reviewer or editor, if any. Looking forward to contributing to the upcoming issue of IJMS.

Sincerely,

Prof Dong-Kug Choi

Head of the Functional Genomics Laboratory and Department of Biotechnology

Director of the Research Institute of Inflammatory Disease (RID)

College of Biomedical and Health Science

Konkuk University, Chungju-27478

South Korea

[email protected];

Tel.: +82-43-840-3610; Fax: +82-43-840-3872

Reviewer 2 Report

The manuscript: Dioscorea nipponica Makino rhizome extract and its active compound dioscin protect against neuroinflammation and scopolamine-induced memory deficits, evaluate the anti-inflammatory effects of DIOSCIN. Besides, the authors have studied the memory-enhancing effects of DNRE, pretreating the animals with scopolamine. Regarding the percentage of cell viability and NO release, it seems that the treatment with 100 micrograms of DNRE has the best effect on cell survival. Still, when they evaluate NO release, it looks like that the best dose is 400 micrograms. In my opinion, there may be some incongruence that the authors should explain. Why are the two best doses significantly different? Four hundred micrograms are less protective than 100 micrograms. In addition, the dose of 100 micrograms plus LPS results is more efficacious than other doses. Here the authors didn't use the 400 micrograms. Why?

Scopolamine, as reported, is an anticholinergic drug. It has a reversible effect. How do the authors explain the "regenerative" effect of DNRE on scopolamine-treated animals? Do they think that it has a cholinergic effect that blocks the scopolamine effects? In my opinion, they should propose a mechanism of action!

Last but not least, the authors should prepare histological and immunohistological analyses to explain which cortical and hippocampal areas are more affected by DNRE. 

Author Response

Title:               Dioscorea nipponica Makino rhizome extract and its active compound dioscin protect against neuroinflammation and scopolamine-induced memory deficits

Journal:          IJMS

Mans. ID        ijms-1823621

Editor

We thank you and reviewer for your and their important time and attention towards our submission. It was your valuable and insightful comments that led to possible improvements of our manuscript in the current version. All authors have considered comments raised by the reviewers and tried their best to address them to the point. We hope that our revised manuscript would meet your high standards. Below you could find authors point-by-point response to the reviewer’s comments. We have indicated all the changes in manuscript as “track change” for the understanding and quick finding.

Reviewer 2:

  1. The manuscript: Dioscorea nipponica Makino rhizome extract and its active compound dioscin protect against neuroinflammation and scopolamine-induced memory deficits, evaluate the anti-inflammatory effects of DIOSCIN. Besides, the authors have studied the memory-enhancing effects of DNRE, pretreating the animals with scopolamine. Regarding the percentage of cell viability and NO release, it seems that the treatment with 100 micrograms of DNRE has the best effect on cell survival. Still, when they evaluate NO release, it looks like that the best dose is 400 micrograms. In my opinion, there may be some incongruence that the authors should explain. Why are the two best doses significantly different? Four hundred micrograms are less protective than 100 micrograms. In addition, the dose of 100 micrograms plus LPS results is more efficacious than other doses. Here the authors didn't use the 400 micrograms. Why?

Response: Indeed, DNRE 200 - 400 µg reduced NO release substantially than 100 µg (Fig 2 B). However, DNRE 200 - 400 µg did not improve cell viability in MTT assay (Fig 2 A), indicating possible toxicity at these doses. We have described this issue in the result section of our revised manuscript.

Based on the cell viability assay we confirmed 100 µg would be suitable higher dose (IC50 = 152 µg/ml of MTT assay), so, we did not use 400 µg concentration in antioxidant assay (Fig 2 C).

  1. Scopolamine, as reported, is an anticholinergic drug. It has a reversible effect. How do the authors explain the "regenerative" effect of DNRE on scopolamine-treated animals? Do they think that it has a cholinergic effect that blocks the scopolamine effects? In my opinion, they should propose a mechanism of action!

Response: Agree with reviewer that at right dose scopolamine has clinical application as anti-cholinergic drug in nausea, constipation and dry mouth. However, this drug is a potential hallucinogenic alkaloid and neurotoxin. Recent development proposed scopolamine could induce Alzheimer’s disease like model in animals via multiple mechanisms [1], for example, cholinergic dysfunction and upregulation of AChE, neuroinflammation and oxidative stress [2-4]. From our study, we proposed dioscin inhibiting scopolamine-induced neuroinflammation (Fig 7 A) in mouse cerebral cortex and hippocampus. This result is well supported by our in vitro anti-inflammatory effect of dioscin (Fig 5 C).

Since neuroinflammation and oxidative stress is one of the hallmarks of neurodegeneration, compound’s ability to ameliorate these two parameters indicate ‘regenerative’ abilities. In this context, our present findings (Fig 7 B) and previous result [5] clarifies that dioscin could increase the neurotrophic factors BDNF and phosphorylated CREB, and spatial learning memory (Fig 6 A-C). Which further justifies the ‘regenerative’ activity.

  1. Last but not least, the authors should prepare histological and immunohistological analyses to explain which cortical and hippocampal areas are more affected by DNRE.

Response: Appreciate this insightful suggestion. The current study has done protein analysis of cerebral cortex and hippocampus regions of mouse brain. And our data indicating that dioscin might have affected cerebral cortex region more efficiently than hippocampus region (Fig 7 A & B). We appreciate reviewer’s suggestion and have addressed this issue in result section of our revised manuscript.

We appreciate and understand that all these comments were raised to improve our manuscript's quality and readability. The authors have attempted their best to address all comments, and we welcome further enquiries by the reviewer or editor, if any. Looking forward to contributing to the upcoming issue of IJMS.

Sincerely,

Prof Dong-Kug Choi

Head of the Functional Genomics Laboratory and Department of Biotechnology

Director of the Research Institute of Inflammatory Disease (RID)

College of Biomedical and Health Science

Konkuk University, Chungju-27478

South Korea

[email protected];

Tel.: +82-43-840-3610; Fax: +82-43-840-3872

References

  1. Chen, W.N.; Yeong, K.Y. Scopolamine, a Toxin-Induced Experimental Model, Used for Research in Alzheimer's Disease. CNS & neurological disorders drug targets 2020, 19, 85-93, doi:10.2174/1871527319666200214104331.
  2. Iqbal, S.; Shah, F.A.; Naeem, K.; Nadeem, H.; Sarwar, S.; Ashraf, Z.; Imran, M.; Khan, T.; Anwar, T.; Li, S. Succinamide Derivatives Ameliorate Neuroinflammation and Oxidative Stress in Scopolamine-Induced Neurodegeneration. Biomolecules 2020, 10, doi:10.3390/biom10030443.
  3. Cheon, S.Y.; Koo, B.N.; Kim, S.Y.; Kam, E.H.; Nam, J.; Kim, E.J. Scopolamine promotes neuroinflammation and delirium-like neuropsychiatric disorder in mice. Scientific reports 2021, 11, 8376, doi:10.1038/s41598-021-87790-y.
  4. Karthivashan, G.; Park, S.-Y.; Kweon, M.-H.; Kim, J.; Haque, M.E.; Cho, D.-Y.; Kim, I.-S.; Cho, E.-A.; Ganesan, P.; Choi, D.-K. Ameliorative potential of desalted Salicornia europaea L. extract in multifaceted Alzheimer’s-like scopolamine-induced amnesic mice model. Scientific reports 2018, 8, 7174, doi:10.1038/s41598-018-25381-0.
  5. Azam, S.; Haque, M.E.; Cho, D.Y.; Kim, J.S.; Jakaria, M.; Kim, I.S.; Choi, D.K. Dioscin-Mediated Autophagy Alleviates MPP(+)-Induced Neuronal Degeneration: An In Vitro Parkinson's Disease Model. Molecules (Basel, Switzerland) 2022, 27, doi:10.3390/molecules27092827.

Reviewer 3 Report

Dear Author,

The manuscript is an elaborate, well conducted study who fully deserve the publication. However, before publication I suggest improving the English quality in the especially in the results chapter. Not only the style needs improvement, but some phrases are Grammarly incorrect. All the text should be seriously revised. Interestingly, some parts of the discussions are well written, but discussions require revision as well.

I would like to have some punctual suggestion I hope you will find them useful.  

Row 89 please indicate the source for the BV-2 cells. It does not make sense to indicate a bibliographical source for it.

Row 164 please use the international abbreviation “g” (not gm)

Row 181 please indicate the bibliographical reference for the Morris water maze.

Row 217 I suggest using Standard deviation instead the SEM…  for this type of studies is more relevant. The methods used for determining IC50 values of DNRE (fig 2 row 232) should be mention in statistics chapter or elsewhere in the methods. IC 50 should be also determined for the MTT assay.

Row 231 figures A and B from figure 2 are not clear… Please find a more intuitive way of presenting these results. Same remark for figures 3 and 4. The figures should be self-explanatory. Please explain the abbreviations. For figure 3 please reconsider the title… IL 1, 6, TNF are not enzymes… Why not presenting them similar to fig 6?

Author Response

Title:               Dioscorea nipponica Makino rhizome extract and its active compound dioscin protect against neuroinflammation and scopolamine-induced memory deficits

Journal:          IJMS

Mans. ID        ijms-1823621

Editor

We thank you and reviewer for your and their important time and attention towards our submission. It was your valuable and insightful comments that led to possible improvements of our manuscript in the current version. All authors have considered comments raised by the reviewers and tried their best to address them to the point. We hope that our revised manuscript would meet your high standards. Below you could find authors point-by-point response to the reviewer’s comments. We have indicated all the changes in manuscript as “track change” for the understanding and quick finding.

Reviewer 3:

Dear Author,

The manuscript is an elaborate, well conducted study who fully deserve the publication. However, before publication I suggest improving the English quality in the especially in the results chapter. Not only the style needs improvement, but some phrases are Grammarly incorrect. All the text should be seriously revised. Interestingly, some parts of the discussions are well written, but discussions require revision as well.

Response: Thank you for this generous suggestion, we have thoroughly read and attempted to correct linguistic and grammar issues according to British/Australian style. We hope reviewed version would be up to the standard.

I would like to have some punctual suggestion I hope you will find them useful.  

Response: Thank you for your generous suggestions, surely, they are useful for the improvement of our manuscript.

Row 89 please indicate the source for the BV-2 cells. It does not make sense to indicate a bibliographical source for it.

Response: Sincere apology for overlooking this fact, we have now added the details of BV-2 cells source.

Row 164 please use the international abbreviation “g” (not gm)

Response: Appreciate this suggestion, we have corrected as per indicated.

Row 181 please indicate the bibliographical reference for the Morris water maze.

Response: Appreciate this suggestion, we have added the MWM reference which we have followed.

Row 217 I suggest using Standard deviation instead the SEM…  for this type of studies is more relevant. The methods used for determining IC50 values of DNRE (fig 2 row 232) should be mention in statistics chapter or elsewhere in the methods. IC 50 should be also determined for the MTT assay.

Response: Appreciate this generous suggestion, we have mentioned IC50 calculation details in the figure legend section. Also, we have added the IC50 value of MTT assay, as indicated, in the figure legend section of Fig 2.

Row 231 figures A and B from figure 2 are not clear… Please find a more intuitive way of presenting these results. Same remark for figures 3 and 4. The figures should be self-explanatory. Please explain the abbreviations. For figure 3 please reconsider the title… IL 1, 6, TNF are not enzymes… Why not presenting them similar to fig 6?

Response: Appreciate this suggestion, we have modified all images indicated and presented more clearly. We have corrected Fig 3 legends with appropriate word.

We appreciate and understand that all these comments were raised to improve our manuscript's quality and readability. The authors have attempted their best to address all comments, and we welcome further enquiries by the reviewer or editor, if any. Looking forward to contributing to the upcoming issue of IJMS.

Sincerely,

Prof Dong-Kug Choi

Head of the Functional Genomics Laboratory and Department of Biotechnology

Director of the Research Institute of Inflammatory Disease (RID)

College of Biomedical and Health Science

Konkuk University, Chungju-27478

South Korea

[email protected];

Tel.: +82-43-840-3610; Fax: +82-43-840-3872

Round 2

Reviewer 3 Report

Dear Author, 

I found a significant improvement in English quality and style. I still have one more suggestion left. Why some pictures  are in colour while  the others are still left in black and white. I suggest finding a unitary pattern for pictures.